# *BvcZR3* and *BvHs1^pro-1^* Genes Pyramiding Enhanced Beet Cyst Nematode (*Heterodera schachtii* Schm.) Resistance in Oilseed Rape (*Brassica napus* L.)

**DOI:** 10.3390/ijms20071740

**Published:** 2019-04-08

**Authors:** Xuanbo Zhong, Qizheng Zhou, Nan Cui, Daguang Cai, Guixiang Tang

**Affiliations:** 1Zhejiang Provincial Key Laboratory of Crop Genetic Resources, Institute of Crop Science, Zhejiang University, Hangzhou 310058, Zhejiang, China; 21716135@zju.edu.cn (X.Z.); 21716136@zju.edu.cn (Q.Z.); 21816113@zju.edu.cn (N.C.); 2Department of Molecular Phytopathology, Christian-Albrechts-University of Kiel, Hermann Rodewald Str. 9, D-24118 Kiel, Germany; dcai@phytomed.uni-kiel.de

**Keywords:** *Hs1^pro-1^*, *cZR3*, gene pyramiding, *Heterodera schachtii*, resistance

## Abstract

Beet cyst nematode (*Heterodera schachtii* Schm.) is one of the most damaging pests in sugar beet growing areas around the world. The *Hs1^pro-1^* and *cZR3* genes confer resistance to the beet cyst nematode, and both were cloned from sugar beet translocation line (A906001). The translocation line carried the locus from *B. procumbens* chromosome 1 including *Hs1^pro-1^* gene and resistance gene analogs (RGA), which confer resistance to *Heterodera schachtii*. In this research, *BvHs1^pro-1^* and *BvcZR3* genes were transferred into oilseed rape to obtain different transgenic lines by *A. tumefaciens* mediated transformation method. The *cZR3Hs1^pro-1^* gene was pyramided into the same plants by crossing homozygous *cZR3* and *Hs1^pro-1^* plants to identify the function and interaction of *cZR3* and *Hs1^pro-1^* genes. In vitro and in vivo cyst nematode resistance tests showed that *cZR3* and *Hs1^pro-1^* plants could be infested by beet cyst nematode (BCN) juveniles, however a large fraction of penetrated nematode juveniles was not able to develop normally and stagnated in roots of transgenic plants, consequently resulting in a significant reduction in the number of developed nematode females. A higher efficiency in inhibition of nematode females was observed in plants expressing pyramiding genes than in those only expressing a single gene. Molecular analysis demonstrated that *BvHs1^pro-1^* and *BvcZR3* gene expressions in oilseed rape constitutively activated transcription of plant-defense related genes such as *NPR1* (non-expresser of *PR1*), *SGT1b* (enhanced disease resistance 1) and *RAR1* (suppressor of the *G2* allele of *skp1*). Transcript of *NPR1* gene in transgenic *cZR3* and *Hs1^pro-1^* plants were slightly up-regulated, while its expression was considerably enhanced in *cZR3Hs1^pro-1^* gene pyramiding plants. The expression of *EDS1* gene did not change significantly among transgenic *cZR3*, *Hs1^pro-1^* and *cZR3Hs1^pro-1^* gene pyramiding plants and wild type. The expression of *SGT1b* gene was slightly up-regulated in transgenic *cZR3* and *Hs1^pro-1^* plants compared with the wild type, however, its expression was not changed in *cZR3Hs1^pro-1^* gene pyramiding plant and had no interaction effect. *RAR1* gene expression was significantly up-regulated in transgenic *cZR3* and *cZR3Hs1^pro-1^* genes pyramiding plants, but almost no expression was found in *Hs1^pro-1^* transgenic plants. These results show that nematode resistance genes from sugar beet were functional in oilseed rape and conferred BCN resistance by activation of a CC-NBS-LRR R gene mediated resistance response. The gene pyramiding had enhanced resistance, thus offering a novel approach for the BCN control by preventing the propagation of BCN in oilseed rape. The transgenic oilseed rape could be used as a trap crop to offer an alternative method for beet cyst nematode control.

## 1. Introduction

Beet cyst nematode (*Heterodera schachtii* Schm.) is an important pest of sugar beet that can cause significant reductions in yield. Unlike most cyst nematode, the beet cyst nematode (BCN) has a wide host range and can infect more than 218 [1] plant species, including family Brassicaceae and Chenopodiaceae such as sugar beet (*Beta vulgaris* L.), oilseed rape (*Brassica napus* L.), and spinach (*Spinacia oleracear*) [2]. In general, nematodes can be controlled by treating with nematicides, growing resistant cultivars, and crop rotation with non-host or trap crops [3]. However, control via nematicides application is difficult and expensive because nematode larvae and eggs are well protected. In addition, the use of nematicides increases the threat of environmental pollution and some are prohibited worldwide. Growing resistant cultivars with a monogenic resistance can induce the emergence of more virulent nematode pathotypes. Oilseed rape (*Brassica napus* L.) is a good host for BCN and always rotates with sugar beet in the agro-farming system. Therefore, breeding for BCN resistant oilseed rape is of great value and importance [4].

A previous investigation on 111 *Brassica* germplasm lines revealed that all lines are susceptible to *H. schachtii* [5]. BCN resistance is found only in a few spices of *Brassiceae* including oil radish (*Raphanussativus* L. *ssp.oleiferus* DC.) and white mustard [6] (*Sinapisalba* L.). Oil radish shows complete resistance and is often used as a trap crop to mitigate the degree of damage in infested fields [2]. Resistant/trap crop could stimulate the hatching of larvae, which invaded the roots, but prevents larvae from fulfilling their life cycle and thus lowers BCN populations. A dominant nematode gene *Hs1^Rph^*, which is located in the radish chromosome *d*, confers beet cyst nematode resistance [7]. Efforts have been made to transfer this resistance gene into oilseed rape genome by intergeneric hybridization [4,6,8,9,10]. However, expression of the resistance gene in a hybrid species remains a great challenge due to the extremely low and unstable inheritance [7]. Cloning of the cyst nematode resistance genes and their subsequent transfer into the rapeseed could be one of the most feasible strategies to induce nematode resistance in oilseed rape.

*Hs1^pro-1^* gene is the first beet cyst nematode resistance gene cloned from sugar beet translocation line (A906001) by a map-based cloning strategy [11]. The translocation line carried the locus from *B. procumbens* chromosome 1 that confers resistance to *H. schachtii* in sugar beet (*Beta vulgaris* L.) [11]. The *Hs1^pro-1^* gene is different from other nematode resistance genes amid the presence of the NBS-LRR structure, such as *Mi* [12], *Gpa2* [13] and *Hero* [14], due to the presence of the NBS-LRR structure. The resistance mechanism of *Hs1^pro-1^* gene is based on the gene-for-gene relationship [15]. The transcript of *Hs1^pro-1^* gene is upregulated about fourfold after one day of nematode infection. However, no considerable change in the transcript accumulation of *Hs1^pro-1^* is recorded in uninfected roots of resistant beet plants [15]. McLean et al. [16] published a complete sequence of the *Hs1^pro-1^* protein, which included an additional 176 amino acid at N-terminal extension conferred resistance to soybean (*Glycine max* L.) cyst nematode (*Heterodera glycines*).

Hunger et al. [17] cloned 47 resistance gene analogs (RGAs) from genomic DNA of sugar beet. Most cloned resistance genes belong to nucleotide-binding site leucine-rich repeat (NBS-LRR) gene family [18,19]. Members of this family have been isolated in both dicot and monocot plants, exhibiting resistance to a variety of plant pathogens, including bacteria, fungi, viruses, and nematodes [20]. The cloning of conserved sequence of NBS domain has been successfully applied to isolate resistance gene candidates or resistance gene analogs (RGAs) from plant genomes, using a degenerate primer-based PCR strategy. For example, from sugar beet translocation line (A906001), *cZR3* was cloned using a degenerated primer-based PCR strategy [21,22]. The gene is similar to a subset of CC-NBS-LRR resistance proteins, including nematode resistance genes *Mi* from tomato and *Gpa2* from potato [12,21]. The phylogenetic analysis shows that this gene originates from the ancestral gene from which *I2C1* [23] (vascular wilt disease resistance), *Xa1* [24] (bacterial blight resistance) and *Cre3* [25] (nematode resistance) were also originated. For CC-NBS domains, RGAs are required for initiation of a necrotic hypersensitive response, as well as for cell death on the initial infection sites [26,27]. RGAs are ubiquitous in plant genomes, and often cluster with close linkage to active resistance genes [28,29]. Because no complete resistance could be observed so far by transgenic sugar beet plants (Cai unpublished data), it is proposed that additional genes are required to confer full resistance [30]. We hypothesized that the *cZR3* gene may interact with the *Hs1^pro-1^* gene to confer resistance against sugar beet nematode in oilseed rape. Therefore, we transferred the *Hs1^pro-1^* and *cZR3* genes into oilseed rape using hypocotyl explants by *A. tumefaciens* mediated transformation method and pyramided the *cZR3Hs1^pro-1^* genes by crossing homozygous transgenic *cZR3* and *Hs1* plants. The beet cyst nematode resistance tests in vitro and in vivo were detected in *cZR3*, *Hs1^pro-1^* and *cZR3 Hs1^pro-1^* pyramiding plants. Possible resistance mechanisms of *Hs1^pro-1^*, *cZR3* and *cZR3Hs1^pro-1^* gene pyramiding were also discussed in the present study.

## 2. Results

### 2.1. Generation of Transgenic Plants and Identification

The recombinant plasmid-DNA carrying *Hs1^pro-1^ and cZR3* genes (Figure 1A,B) were used for transformation to generate transgenic *Hs1^pro-1^* and *cZR3* plants. Figure 1C–F shows the process of transgenic *Hs1^pro-1^* and *cZR3* plants generation. The hypocotyl segments were pre-cultured on the CIM to induce callus formation for two days and then infected with *A. tumefaciens*. After co-cultivation, explants were cultivated on SIM (Figure 1C) supplemented with 500 mg/L Carb to eliminate the extra bacterium and 50 mg/L Kan to select the positive transgenic plants. The adventitious shoots were observed from the cut end of hypocotyl explants 30 days after cultivation in SIM (Figure 1D). The adventitious shoots were cut off and transferred to shoot elongation medium supplemented with 500 mg/L Carb and 50 mg/L Kan. The putative green transgenic shoots were observed 30 after cultivation in SEM (Figure 1E). The green shoots were transferred to rooting medium to develop the roots (Figure 1F). The rooted plantlets were transferred to the soil and the survived putative transgenic plants were assayed by PCR (Figure 1G,H) and Southern blot (Figure 1I). In general, 194 explants were transformed with pAM194-Hs1 vector and four rooting seedlings were obtained, of which three were identified as transgenic seedlings by PCR (Figure 1G) and Southern blot analysis (Figure 1I). A total of 162 explants were transformed with pAM194-cZR3 vector and 12 rooting seedlings were obtained, of which three were identified as transgenic seedlings by PCR (Figure 1H) and Southern blot analysis (Figure 1I).

### 2.2. Generation of cZR3Hs1^pro-1^ Gene Pyramiding Plants

T1 seeds were obtained by self-pollination of the six T0 independent transgenic *Hs1^pro-1^* (*Hs1^pro-1^*-1, -2, -3) and *cZR3* (*cZR3*-1, -2, -3) oilseed rape plants. The segregation ratio of T1 progeny was determined by PCR analysis using *Hs1^pro-1^* and *cZR3* genes specific primers. The exogenous gene began to segregate in T1 progeny and could be inherited to the next generation (Table 1). The results show the segregation ratio of *Hs1^pro-1^*-1, -3 and *cZR3*-1, -2, -3 were nearly 3:1 and the χ^c2^ test was positive.

*Hs1^pro-1^ cZR3* genes pyramiding plants were generated by artificial hybridization using *cZR3* plant as female and *Hs1^pro-1^* plants as male. Five cross combinations carried out between *cZR3* and *Hs1^pro-1^* independent transgenic lines. In total, 20, 15, 27, 14 and 35 seeds were harvested for F1 *cZR3*-1 × *Hs1^pro-1^*-1-2, *cZR3*-2.1 × *Hs1^pro-1^*-1-3, *cZR3*-2.2 × *Hs1^pro-1^*-1-3, *cZR3*-3 × *Hs1^pro-1^*-1-2 and *cZR3*-3 × *Hs1^pro-1^*-3, respectively (Table 2). Segregation of F2 crossed progeny occurred and the heterozygous *cZR3Hs1^pro-1^* gene pyramiding was identified by PCR analysis using specific gene primers (Figure 2). There were four possible exogenous gene combinations: e.g., Type I: plants, containing both *Hs1^pro-1^* and *cZR3* genes, Type II and III: plants including a single exogenous gene *Hs1^pro-1^* or *cZR-3* and Type IV: plants including neither *Hs1^pro-1^* nor *cZR3*. Of all 111 F2 individuals, we identified 32 with *cZR3*-plants and 19 with *Hs1^pro-1^*- plants, 47 with *cZR3Hs1^pro-1^* genes pyramiding plants, and 13 neither *Hs1^pro-1^* nor *cZR3* genes. The segregation ratio of F2 cross progeny was 47:32:19:13 (9:3:3:1) which were consistent with Mendel’s laws of inheritance.

### 2.3. Beet Cyst Nematode Test In Vitro and In Vivo

To determine the resistance to BCN of the *cZR3* and *Hs1* genes and *cZR3Hs1* gene pyramiding in oilseed rape, BCN resistance tests in vitro and in vivo were performed in T2 *cZR3*, *Hs1^pro-1^* and F3 *cZR3Hs1^pro-1^* pyramiding seeds. For this, transgenic and gene pyramiding seeds were geminated on agar plates containing 150 mg/L kanamycin for selection of transgenic plants (Figure 3). The surviving seedlings were transferred to six-well plates for nematode infection experiments (Figure 4A), where each plant was inoculated with 200 infective nematode juveniles and was repeated three times for individual plant. The J2 penetration rate was determined one week after nematode inoculation. On average 15–22% of inoculated BCN J2 juveniles penetrated the plants containing exogenous genes as well as the wild type plants. However, a significant difference was observed between the transgenic and the control plants by counting developed females six weeks after nematode inoculation. Most juveniles in transgenic plants were not fully developed and became smaller and translucent, whereas the well-developed nematode females were easily recognizable on wild type plants (Figure 4B). These results demonstrate that *Hs1^pro-1^* and *cZR3* could confer a certain level of resistance to BCN in oilseed rape. There was a significant difference in the reduction of the number of developed females among transgenic generations and *cZR3Hs1^pro-1^* gene pyramiding plants. On average, 10.8 ± 0.9 developed females were found in each wild type plant, while 5.3 ± 1.4 developed females in *cZR3*, 7.1 ± 2.4 in *Hs1^pro-1^*, and 4.1 ± 1.3 in *cZR3:Hs1^pro-1^* pyramiding plants were counted (Figure 5). Thus, *cZR3Hs1^pro-1^* gene pyramiding could enhance the BCN resistance by decreasing the number of developed females per plant.

BCN resistance tests in vivo were performed under the greenhouse condition. Six weeks after BCN inoculation, the number of developed females per plant was counted. The number of developed females in wild type was 50 ± 4.7, whereas the number of developed females among *cZR3*, *Hs1^pro-1^* and *cZR3Hs1^pro-1^* was 38 ± 7.4, 33 ± 3.7 and 24 ± 5.2, respectively (Figure 6). There was a significant difference in the number of developed females among wild type, *cZR3* and *Hs1^pro-1^* transgenic and *cZR3Hs1^pro-1^* pyramiding plants. 

### 2.4. Determination of RGA-Mediated Signaling Pathways

To clarify whether expression of *cZR3* and *Hs1^pro-1^* activates a specific signaling pathway, the transcript levels of four key genes involve in distinct defense pathways were analyzed in transgenic *cZR3*, *Hs1* and *cZR3Hs1* gene pyramiding plants compared with those in wild type by semi-RT-PCR (Figure 7). The gene list included *NPR1* (non-expresser of *PR1*) and *EDS1* (enhanced disease resistance 1), which are key regulators of resistance responses triggered by TIR-NBS-LRR-R-proteins mediated response and *SGT1* (suppressor of the *G2* allele of *skp1*) as well as *RAR1* (required for *Mla12* resistance), which are both involved in the non-TIR-(CC) NBS-LRR-R proteins mediated responses [31]. The expression of *NPR1* gene in transgenic *cZR3* and *Hs1^pro-1^* plants seemed to be slightly up-regulated and there was enhanced expression in *cZR3Hs1^pro-1^* gene pyramiding plants. The expression of *EDS1* gene did not change significantly among transgenic *cZR3* and *Hs1* plants, *cZR3Hs1^pro-1^* gene pyramiding plants and wild type. The expression of *SGT1b* gene was slightly up-regulated in transgenic and *Hs1^pro-1^* plants compared with the wild type, however, its response in *cZR3Hs1^pro-1^* gene pyramiding plant was similar to the wild type and had no interaction effect. *RAR1* gene expression was significantly up-regulated in *cZR3* transgenic plants, but almost no change in expression was found in *Hs1^pro-1^* transgenic plants compared with wild type plants. However, *RAR1* gene expression was also significantly enhanced in *cZR3Hs1^pro-1^* gene pyramiding plants. Furthermore, the relative expression of *SGT1b* and *RAR1* genes was also analyzed. Compared with wild type, expression level of *SGT1b* in *cZR3* transgenic plants and *cZR3Hs1^pro-1^* pyramiding plants was decreased slightly, while it was increased in *Hs1* transgenic plants (Figure 8A). The expression of *RAR1* gene by qRT-PCR analysis showed that *RAR1* was slightly expressed in wild type and *Hs1^pro-1^* transgenic plants. The expression of *RAR1* in *cZR3* transgenic plant was about 50 times higher than the wild type, while the expression of *RAR1* was enhanced by *cZR3Hs1^pro-1^*gene pyramiding, which was about 150 times higher than the wild type (Figure 8B).

## 3. Discussion

Crops are attacked by nematodes causing considerable economic losses worldwide. The estimated worldwide losses due to plant parasitic nematodes are about $125 billion annually [32]. An integrated strategy, e.g., including trap crop planting, nematicide application and cultivation of resistant sugar beet varieties, could control beet cyst nematode infestation in sugar beets [33]. Effective nematode control could be achieved by cultivating a nematode resistant variety that can reduce nematode populations up to 70% in field conditions [33]. Breeding of resistant cultivars is the most desired and promising alternative because regeneration of whole transgenic sugar beet plants with *A. tumefaciens* mediated transformation is laborious and timing consuming. In the present study, we transferred *BvHs1^pro-1^* and *BvcZR3* resistance genes, into oilseed rape to develop cyst nematode resistance crop. In vitro and in vivo nematode tests showed the developed female reduced 50.93%, 34.26%, 24.0%, and 34.0%, respectively, in *BvcZR3* and *BvHs1^pro-1^* transgenic plants. Our results are consistent with previous studies, where *A. rhizogenes*-mediated transformation was performed to generate transgenic *Hs1^pro-1^* and *cZR3* hairy roots in sugar beet [11,21].

Different R genes often confer resistance to different isolates, races or biotypes of beet cyst nematode. The simultaneous expression of different R genes could broaden the resistance spectrum, as it may provide resistant against various races or isolates [34]. The multiple genes pyramiding strategy has been applied in genetic engineering to achieve durable resistance against phytopathogen [35,36] and nematode resistance [37,38]. Bharathia et al. [36] reported *Allium sativum* (*asal*) and *Galanthus nivalis* (*gna*) lectin genes pyramided into rice lines through sexual crosses between two stable transgenic rice lines, which endowed the pyramided rice lines with enhanced resistance to major sap sucking insects. Urwin et al. [38] transferred cowpea trypsin inhibitor (*CpTI*) and a cystatin gene (*Oc-ID86*) into *Arabidopsis* and the *CpTI Oc-ID86* transgenic plants displayed enhanced resistance against *H. schachtii*. Chan et al. [37] demonstrated via dual gene overexpression system that utilizing a plant cysteine proteinase inhibitor (*CeCPI*) and a fungal chitinase (*PjCHI-1*) in tomato (*Solanum lycopersicum*) can enhance resistance against root-knot nematode (*Meloidogyne incognita*). The *Hs1^pro-1^* gene encodes a plasma membrane protein with an extensive leucine-rich region, which contains a transmembrane spanning domain and a short hydrophobic C-terminal domain. It could be speculated that *Hs1^pro-1^* resides in the plasma membrane as a receptor with its N-terminus toward the extracellular space [11]. Similar to recently cloned nematode R-genes, including *Mi* [12] and *Gpa2* [21], *cZR3* belongs to the CC-NBS-LRR class of R-protein lacking a signal sequence [21]. Previous studies show that no complete resistance is observed in *Hs1^pro-1^* transgenic sugar beet plants and a second gene is proposed to be essential for the resistance [30]. In this study, *cZR3Hs1^pro-1^* genes pyramiding plants were generated through sexual crosses made between two transgenic *Hs1^pro-1^* and *cZR3* plants of oilseed rape. Cyst nematode resistance test in vitro and in vivo showed that the number of females that developed per plant on transgenic *cZR3Hs1^pro-1^* genes pyramiding plants were reduced 52.0% and 62.0%, respectively, and the reduction percentage was significantly different with transgenic *cZR3* and *Hs1^pro-1^* and wild type plants. The possible functional model for *cZR3Hs1^pro-1^* mediated response was based on *cZR3,* which may function as a co-receptor with *Hs1^pro-1^* together recognizing the *Avr* products released by nematodes into the cytoplasm initiating signal transduction that finally leads to resistance response.

Thus far, little is known about the complex regulatory role of cloned nematode R-genes. It is generally believed that these genes recognize nematode effectors and trigger specific signaling pathways that lead to resistance responses [39]. The disease resistance mechanisms in model plant *Arabidopsis thaliana* were extensively studied, and the gene-for-gene hypothesis has been proposed for a long time. R genes interact with the corresponding pathogenic toxic genes, thus causing local reactive oxygen species accumulation, programmed cell death and local allergic reactions [40]. On the one hand, these reactions limit the growth and extend of pathogenic bacteria in infected sites. On the other hand, these reactions release signaling molecules to surrounding cells, further inducing expression of defense genes to boost whole-plant resistance. Some crucial genes mediated by R gene have been found in recent years, such as *RAR1*, *NPR1*, *EDS1* and *SGT1*. Studies have shown that the CC-NBS-LRR R gene mediates disease resistance reaction associated with *RAR1* and *SGT1*, while *NPR1*, *EDS1* and *PAD4* are related with TIR-NBS-LRR R genes [31]. This study proved that cZR-3 and *Hs1^pro-1^* could independently activate a *RAR1/SGT1* dependent signaling pathway in plants, which is essential for a CC-NBS-LRR R gene mediated resistance response. Hence, the present study provides a new approach to develop BCN resistance in oilseed rape plants based on stacking of *cZR3Hs1^pro-1^* genes that confer a high level of BCN resistance in transgenic plants.

## 4. Materials and Methods

### 4.1. Plant Transformation

Direct shoots regeneration from hypocotyl explants of semi-winter-type oilseed rape cultivar *zheshuang 758* were transformed by *A. tumefaciens*, as described by Tang et al. [41]. The PAM194-*Hs1^pro-1^* and PAM194-*cZR3* vector contains the complete *Hs1^pro-1^* ORF (Gene accession number U79733) and *cZR3* (Gene accession number DQ907613), respectively, driven by cauliflower 35S promoter and nopaline synthase terminator, the neomycin phosphotranferase marker gene (*nptII*) and β-glucuronidase (GUS) reporter gene (Figure 1A,B) driven by the cauliflower 35S promoter. Both vectors were presented by Prof. Daguang Cai from Institute of Molecular Phytopathology, Kiel University, Germany. The binary vectors were then transformed into the competent cells of *A. tumefaciens* strain GV3101 by the freeze–thaw method.

In brief, we used about 10 mm hypocotyl segments was obtained by growing seeds of oilseed rape seeds on germination medium [42] (2.22 g/L 1/2MS basal medium from Phytotechnology Laboratories^®^, Shawnee Mission, Lenexa, KS, USA, 20 g/L sucrose, 8 g/L Agar, pH 5.8) at 25 °C and 16 h light/ 8 h dark periods for about 4–5 days as explants. The isolated explants were pre-cultured on the callus induction medium (CIM) (MS 4.43 g/L, 1 mg/L 2,4-D, 30 g/L sucrose, 8 g/L Agar, pH 5.6) at 25 °C and 16 h light/ 8 h dark periods for 3 days. The pre-cultured explants were infected by *A. tumefaciens* suspension (OD600 0.3–0.4) for 10 min, then subsequently blotted on sterile filter paper and placed on CIM at 22 °C under dim light co-cultivation for 3 days. After co-cultivation, the explants were placed on SIM media for shoot generation (SIM) (MS 4.43 g/L, 4 mg/L N-6-benzylaminopurine (BAP), 0.1mg/L 1-Naphthaleneacetic acid (NAA), 5 mg/L silver nitrate (AgNO_3_), 500 mg/L carbenicillin (Carb), 50 mg/L kanamycin (Kan), 30 g/L sucrose, 8 g/L Agar, pH5.8) at 25 °C and 16 h light/ 8 h dark periods for 2 weeks. The media were replaced every two weeks until multiple shoots were generated. The multiple induced shoots were isolated and transferred to shoot elongation media [43] (SEM) (B5 basal medium from Phytotechnology laboratories^®^ 3.21 g/L, 10 g/L sucrose, 50 mg/L Kan, 9 g/L Agar, pH5.8). The elongated shoots were transferred to SEM to develop the roots. The rooting plants were first grown in 10 cm diameter pots containing nutrient soil (Pindstrup, Ryomgaard, Denmark). The surviving plants were transferred to greenhouse to vernalize at 4 °C 16 h light/ 8 h dark periods for 40 days. The vernalized plants were transferred to the greenhouse to grow normally to harvest seeds at 25 °C and 16 h light/ 8 h dark periods. When the plants began to flower, the inflorescence of each independent putative transformed line was covered with transparent plastic bags and the plants were allowed self-pollination to acquire T1 seeds.

### 4.2. PCR Analysis

Forty-day-old rapeseed leaves were used to extract genomic DNA from transgenic and wild-type plants using CTAB method. PCR analyses were performed using specific gene primers for *Hs1^pro-1^* (accession number: U79733.1) (F: 5′-GGCACCATCCAAACTCGG-3′, R: 5′- CGAATAAGTGAGAGGATC-3′), *cZR3* (accession number: DQ907613) (F: 5′- GGCAAAACTGCTCTTGCC-3′ and R: 5′-AGCCCTATCAATAACTCC-3′) and *cZR3* (F: 5′-AGTTATTGATAGGGCTATGG-3′ and R: 5′-ATACTTGAAGCAGTCAGG-3′). And the size of amplifying products are 500-bp [11], 710-bp and 410-bp [22] respectively. The PCR reaction was performed at 94 °C for 3 min, followed by 35 cycles of 94 °C for 1 min, 55 °C (*Hs1^pro-1^* and *cZR3*) for 1 min and 72 °C for 1 min 20 s with a final extension at 72 °C for 10 min. PCR-amplified products were analyzed on 1% agarose gel, stained with ethidium bromide and fluoresce under ultraviolet light. 

### 4.3. Southern Blot Analysis

Genomic DNA of each independent transgenic *cZR3* and *Hs1^pro-1^* lines was restricted by Hind III (Takara Bio Inc., Dalian, China) at 37 °C for 5 h, and was separated on 0.7% agarose gel overnight and transferred onto a Hybond N^+^ nylon membrane (GE Healthcare, RPN 303B, piscataway, NJ, USA) using the alkaline transfer buffer as recommended by the manufacturer. Southern blots were hybridized using 32P-labeled *cZR-3* and *Hs1^pro-1^* DNA fragment as probe at 62 °C overnight. The blots were washed twice with 0.5× saline sodium citrate (SSC), 0.2% *w*/*v* SDS for 30 min and together with the film exposed at −70 °C for 48 h [44].

### 4.4. Histochemical GUS Assays

The histochemical *GUS* assay was carried out according to Jefferson et al.’s [45] method. Ten milliliters of X-Gluc solution including 50 mM Na_3_PO_4_ buffer (pH 7.0), 0.2 mg/mL X-Gluc (5-bromo-4-chloro-3-indolyl β-D-glucuronide) and two drops of Triton-100 were added in the Petri-dish, which contained the putative transgenic leaves and plants. The leaves and plants were covered with X-Gluc solution and incubated at 37 °C for 16 h. Afterwards, the samples were washed using 70% ethanol to remove the chlorophyll. The GUS staining signals were evaluated under a stereomicroscope (Stemi SV 11, Zeiss, Jena, Germany).

### 4.5. Genes Pyramiding and Progeny Analysis

*Hs1^pro-1^*and *cZR3* genes pyramiding were generated by artificial hybridization using T0 putative transgenic *cZR3* line as female and T0 putative transgenic *Hs1^pro-1^* line as male. F1 seeds obtained from the crossed inflorescence and *cZR3Hs1^pro-1^* positive F2 plant were screened for the presence of *cZR3Hs1^pro-1^* genes by PCR.

T1 seeds obtained from self-pollination of 3 T0 independent *Hs1^pro-1^* and *cZR3* plants were sown in 9 cm Petri-dishes containing germination medium with 150 mg/L kanamycin. The surviving plants were transferred to pots containing peat moss (Pindstrup, Balozi, Lativa) and maintained in green house. All T1 were screened for the presence of *Hs1^pro-1^* or *cZR3* genes by PCR. Subsequent generations were obtained by self-pollination of transgenic plants and confirmed by PCR.

### 4.6. Propagation of H. Schactii

In vitro cultured mustard (*Sinapis albacv*. Albatros) was used as host plants for beet cyst nematode (*H. schactii*) stock propagation. The mustard seeds were surface sterilized in 70% ethanol and in 5% NaClO solution, containing 2–3 drops of Tween 20 for 10 and min, then rinsed at least 3 times with sterilized water. The sterilized seeds were sown on half-strength MS media with 0.8% agar on 9 cm Petri-dishes at 25 °C in the dark. After 7-days of germination, the seedlings were transferred to 15 cm Petri-dishes containing 0.2× KNOP medium with 2% (*w*/*v*) sucrose and 0.8% (*w*/*v*) Daishin agar (Duchefa, Haarlem, The Netherlands) and placed in dark for 4 weeks at 25 °C [46]. The cysts were picked from the fields and J2 larvae hatched on 50 μm gauze stimulated by incubating cysts in 3 mM ZnCl_2_ for 7–12 days. The J2 larvae were collected with 10 μm gauze, surface-sterilized with 0.1% HgCl_2_, washed four times in sterile water and re-suspended in 0.2% (*w*/*v*) Gelrite (Duchefa). The sterile J2 larvae were directly inoculated to the mustard roots and the cysts were propagated in the mustard roots in vitro. These plates were used as a stock for beet cyst nematode.

### 4.7. Nematode Resistance Assay In Vitro and In Vivo

T2 transgenic *Hs1^pro-1^*, *cZR3* and F3 *Hs1^pro-1^cZR3* gene pyramiding oilseed rape plants were used for nematode infestation analysis, under both in vitro and in vivo. Nematode resistance analysis in vitro was performed according to Sijmons et al. [47]. First, the T1 transgenic *Hs1^pro-1^* and *cZR3* and F3 *Hs1^pro-1^cZR3* gene pyramiding seeds were surface sterilized, and then germinated on 15 mm Petri dishes containing half-length MS germination medium supplemented with 150 mg/L kanamycin to select the positive transgenic plants. Ten days after germination, the surviving seedlings were transferred to a six-well plate containing KNOP medium and cultured for further 7 days at 23 ± 1 °C with a fluorescent light illumination regime of 16 h/8 h (day/night, 100 µ·mol m^−2^ s^−1^) and the relative humidity was 75%. Two hundred sterile infective J2 larvae were inoculated to each plant by a veterinary syringe. The number of females that developed per plant was counted 6 weeks after inoculation under a stereomicroscope (Stemi SV11, Zeiss, Jena, Germany). At least three 6-well plates were performed for each transgenic line and wild type as biological replicates. For nematode resistance test in vivo, the T2 transgenic *Hs1^pro-1^*, *cZR3* and F3 *Hs1^pro-1^cZR3* genes pyramiding seeds were germinated on the germination medium with 150 mg/L kanamycin for 10 days to select the positive transgenic ones. The positive transgenic single plants were transplanted in plastic tubes (3 cm × 4 cm × 20 cm) filled with silver sand and moistened with Steiner I nutrient solution [48] and cultured in a greenhouse at 25 °C 16 h/8 h (light/dark) periods for further 2 weeks. Each plant was then inoculated with 2 mL suspension containing approximately 2000 pre-hatched J2 BCN larvae by a veterinary syringe. Six weeks after BCN inoculation, the root system was washed free from sand by high-speed tap water and the number of developed BCN cysts was counted. Each line and wild type were replicated 15 plants at least. 

### 4.8. Semi Real Time PCR Analysis

For real time PCR analysis, the total RNA from leaves and roots was extracted with Trizol (Gibco, BRL life technologies, Grand Island, NY, USA) according to the manufacturer’s protocol. First strand cDNA was synthesized with 1 μg of purified total RNA using PrimeScriptTMRT reagent Kit with gDNA Eraser (TaKaRa). Real-time PCR reaction was carried out according to SYBR Premix Ex Taq II (TaKaRa). Specific primers of defensing genes were designed according to sequences of *Arabidopsis* defensing genes in NCBI and summarized in Table 3. The semi-quantitative PCR was performed in 50 µL reactions consisting of 2.5 µL 10 ng/µL cDNA, 5 µL 10× buffer, 0.5 µL 10 mM dNTPs, 5 µL each of 10 pmol/µL primer, 2.5 units of Taq polymerase and 31.5 µL H_2_O under the PCR program: 94 °C for 50 s, 51 °C for 1 min and 72 °C for 1 min for 25 cycles, followed by 10 min at 72 °C. Amplicons were separated on a 1% (*w*/*v*) agarose gel and visualized under UV-light. The levels of gene expression were calculated by comparison of the densities of the PCR products, in which house-keeping *ubiquitin* gene served as an internal control and the mRNA levels for each cDNA probe were normalized to the ubiquitin message RNA level (5’-ACTCTCACCGGAAAGACAATC-3’ and 5’-TGACGTTGTCGATGGTGTCAG-3’). Quantitative RT-PCR was carried out using a SYBR Premix Ex22Taq (perfect real time) kit (TaKaRa Biomedicals) on a LightCycler23480 machine (Roche Diagnostics, Rotkreuz, Switzerland) according to the manufacturer’s instructions (Roche Diagnostics). The qRT-PCR amplification was performed at 94 °C for 10 s, 58 °C for 10 s and 72 °C for 10 s. All reactions were repeated three times. The relative level of gene expression was calculated using the formula 2^−^^△△*CP*^ according to Livak and Schmittgen [49].

### 4.9. Data Analysis

The segregation rates of the T2 progenies of transgenic oilseed plants, as well as the segregation ratios in the F2 generation from crosses between *cZR3* and *Hs1* homozygous transgenic plants were analyzed using the chi-square test to confirm the expected Mendelian segregation pattern of 3:1 (transgenic: non-transgenic plants) and 9:3:3:1. Data were analyzed using IBMSPSS 23.0 statistical system for windows (SPSS Inc., Armonk, NY, USA). Duncan multiple range tests were performed at the 0.05 level of probability.

## Figures and Tables

**Figure 1 ijms-20-01740-f001:**
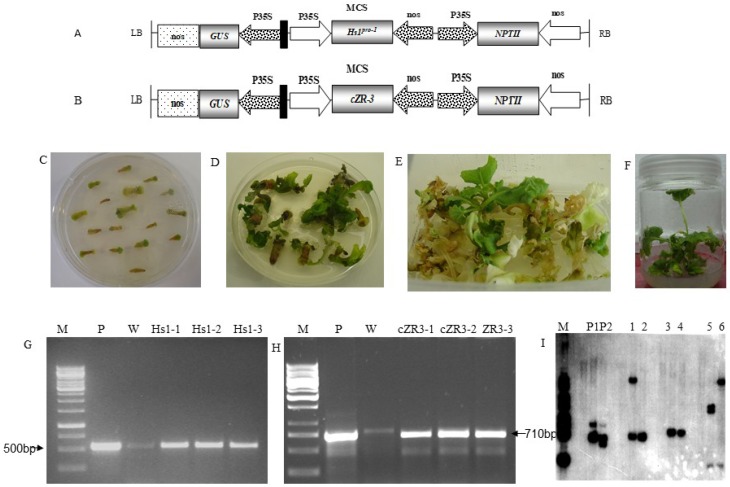
Generation and identification of transgenic *Hs1^pro-1^* and *cZR3* plants from hypocotyle explants of B. *napus* cv. *Zheshuang 758* by *A. tumefaciens* mediated transformation. (**A**,**B**) Diagram of T-DNA region of the binary vector pAM194-*Hs1^pro-1^* (**A**) and pAM194-*cZR3* (**B**). LB, T-DNA left border; RB, right border; P35S, Cauliflower mosaic virus (CaMV) 35S promoter; NOS, nopaline synthase terminator; MCS, multi clone site including Xhol restriction site; *Hs1^pro-1^*, *Beta vulgaris Hs1^pro-1^* gene open read fragment sequence; *cZR3*, *Beta vulgaris* resistance gene sequence; *GUS*, β-glucuronidase report gene; *NPTII*, neomycin phosphotranferase gene for kanamycin resistance. (**C**) Callus induction from hypocotyl explants after co-cultivation with *A. tumefaciens*. (**D**) Shoots regenerated on SIM. (**E**) Shoots elongation on SEM. (**F**) Shoots rooted on RM. (**G**,**H**) PCR assay for transgenic *Hs1^pro-1^* plants (**G**) and *cZR3* plants (**H**) M means 1 kb DNA ladder; P means *Hs1* or *cZR3* plasmid DNA, W means non-transgenic plant DNA; Hs1-1, Hs1-2 and Hs1-3 mean T0 independent transgenic *Hs1^pro-1^* plants; cZR3-1, cZR3-2 and cZR3-3 mean T0 independent transgenic *cZR3* plants. (**I**) Southern blot assay for T0 independent transgenic *cZR3* and *Hs1^pro-1^* lines, P1P2 means *Hs1* and *cZR3* plasmid DNA, respectively; 1, 3, and 5 mean T0 independent transgenic *Hs1* plants; 2, 4, and 6 mean T0 independent transgenic *cZR3* plants.

**Figure 2 ijms-20-01740-f002:**
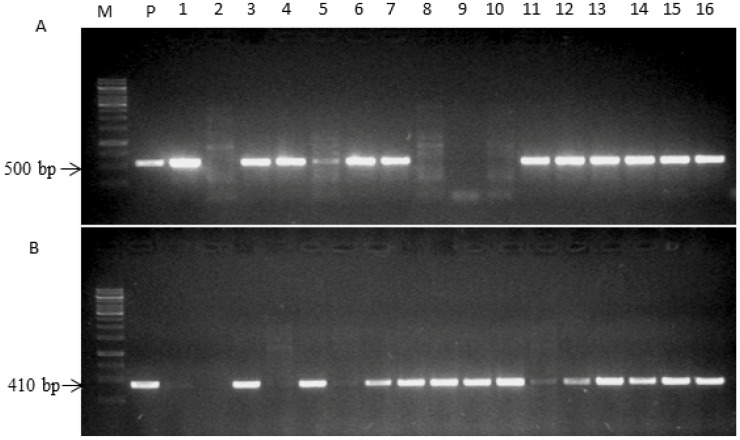
Identification of *cZR3Hs1^pro-1^* pyramiding homozygous plants in F2 crossing progeny by PCR assay using specific *cZR3* and *Hs1^pro-1^* gene primers. The amplification fragment length of *Hs1^pro-1^* and *cZR3* gene was 500 bp and 410 bp, respectively. (**A**) Identification of *Hs1^pro-1^* gene in F2 crossing progeny. (**B**) Identification of *cZR3* gene in F2 crossing progeny. M represents a molecular marker (1 kb); P represents plasmid DNA for *Hs1^pro-1^* or *cZR3* binary vector; 1–16 represent different F2 rapeseed plants.

**Figure 3 ijms-20-01740-f003:**
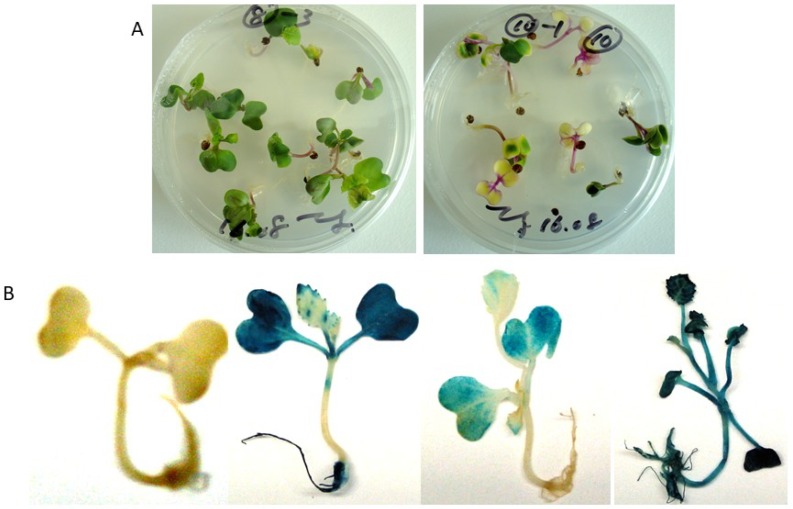
Selection of transgenic rapeseed plants on MS germination containing 150 mg/L kanamycin medium (**A**) and histochemical GUS staining (**B**). The healthy and green plants were used for nematode infestation in vitro and in vivo.

**Figure 4 ijms-20-01740-f004:**
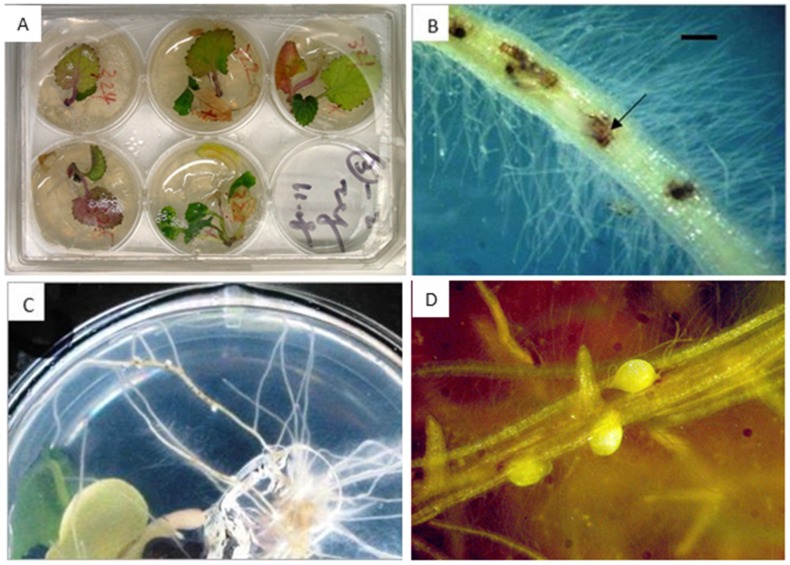
Beet cyst nematode resistance analysis in vitro. (**A**) The kanamycin resistance transgenic plants were planted on the six-well plate with KNOP medium. (**B**) Second stage juveniles (J2s) were inoculated near the plant root and larval penetration could be seen in the oilseed rape roots (dark arrow). The black bar equals 500 μm. (**C**) Female cyst nematode developed on the root surface after 21 days of inoculation with J2. (**D**) The stereo microscope figures of developed female on oilseed rape root.

**Figure 5 ijms-20-01740-f005:**
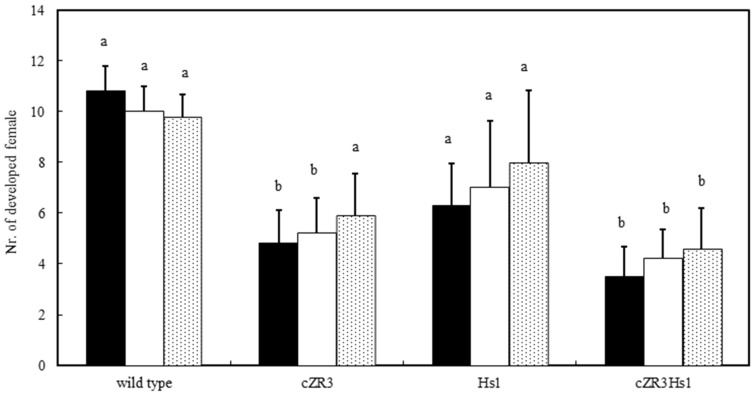
Numbers of developed female per plant on transgenic and wild type plants according to cyst nematode test in vitro. Wild type means non-transgenic oilseed rape (*B. napus* L.); transgenic plants consist of cZR3 and Hs1; and cZR3Hs1 means *cZR3*, *Hs1^pro-1^* and *cZR3Hs1^pro-1^* populations. The black, white, dotted rectangles represent independent wildtype plants and transgenic lines. The averages ± standard errors from three separate replicates are shown. The values with different letters are significantly different at *p* ≤ 0.05 as determined by Duncan’s test (a, b).

**Figure 6 ijms-20-01740-f006:**
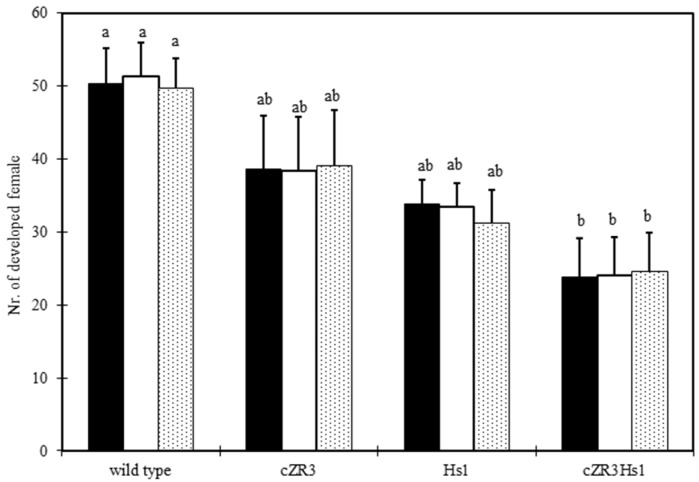
Numbers of developed female per plant on transgenic and wild type plants according to cyst nematode test in vivo. Wild type means non-transgenic oilseed rape (*B. napus* L.); transgenic plants consist of cZR3 and Hs1; and cZR3Hs1 means *cZR3*, *Hs1^pro-1^* and *cZR3Hs1^pro-1^* plants. The black, white, dotted rectangles represent independent wildtype plants and transgenic lines. The averages ± standard errors from three separate replicates are shown. Values with different letters are significantly different at *p* ≤ 0.05 as determined by Duncan’s test (a, b).

**Figure 7 ijms-20-01740-f007:**
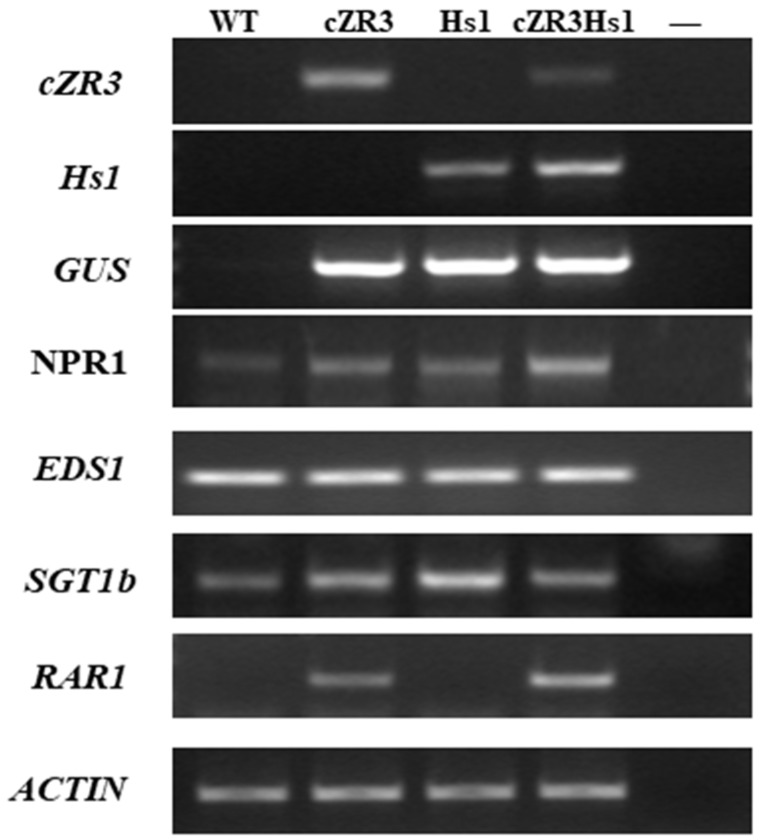
Expression of the key defense related genes in wild-type, transgenic *cZR3* and *Hs1^pro-1^*, and *cZR3Hs1^pro-1^* gene pyramiding plants. The expression levels of *cZR3*, *Hs1^pro-1^*, *GUS*, *NPR1*, *EDS1*, *SGT1* and *RAR1* were determined by semi-quantitative RT-PCR with independent transgenic and pyramiding lines, while oilseed rape wild type *zheshuang 758* served as control. “–“ represents the negative control.

**Figure 8 ijms-20-01740-f008:**
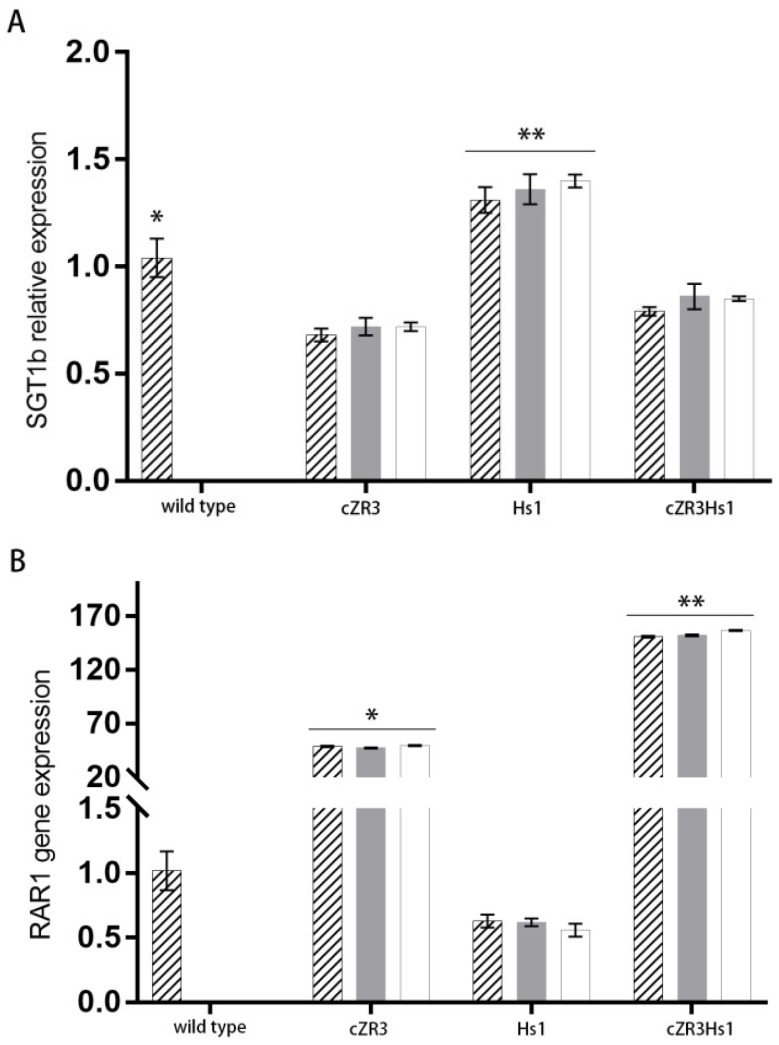
The relative expression of *SGT1b* (**A**) and *RAR1* (**B**) gene among wild type, *cZR3* and *Hs1^pro-1^*, and *cZR3Hs1^pro-1^* pyramiding plants by qRT-PCR. The diagonal stripes, grey, white rectangles represent independent wildtype plants and transgenic lines. The averages ± standard errors from three separate replicates are shown. Significant difference is indicated by different symbols (* and **), as determined by Duncan’s test.

**Table 1 ijms-20-01740-t001:** Segregation rates of T1 progeny of three independent transgenic *Hs1^pro-1^* and *cZR3* lines.

Lines	Number of Plants			Expected Ratio	*p*-Value
	T1	*Hs1^pro-1^* or *cZR3*+	*Hs1^pro-1^* or *cZR3*−		
*Hs1^pro-1^*-1	21	19	2	3:1	*p* > 0.05
*Hs1^pro-1^*-2	6	1	5	3:1	*p* < 0.05
*Hs1^pro-1^*-3	18	16	2	3:1	*p* > 0.05
*cZR-3*-1	8	6	2	3:1	*p* > 0.05
*cZR-3*-2	10	8	2	3:1	*p* > 0.05
*cZR-3*-3	37	27	10	3:1	*p* > 0.05

**Table 2 ijms-20-01740-t002:** Segregation ratios in the F2 generation derived from a cross between homozygous transgenic *Hs1^pro-1^* and *cZR3* plants.

Cross Combination	Number of Plants					Expected Ratio	*p*-Value
	F2	*cZR3Hs1^pro-1^*+	*cZR3* +	*Hs1^pro-1^*+	*cZR3Hs1^pro-1^*−		
*cZR3*-1 ×*Hs1^pro-1^*-2	20	10	3	4	3	9:3:3:1	*p* > 0.05
*cZR3*-2.1 ×*Hs1^pro-1^*-3	15	1	12	1	1	9:3:3:1	*p* < 0.05
*cZR3*-2.2 ×*Hs1^pro-1^*-3	27	16	4	4	3	9:3:3:1	*p* > 0.05
*cZR3*-3 ×*Hs1^pro-1^*-2	14	6	2	2	4	9:3:3:1	*p* < 0.05
*cZR3*-3 ×*Hs1^pro-1^*-3	35	15	8	9	3	9:3:3:1	*p* > 0.05

**Table 3 ijms-20-01740-t003:** Primers were used in the described gene expression of defense pathway.

Target Gene	Accession Number	Primer Sequence (5′ → 3′)	References
*NPR1*	AT1G64280.1	TGAATTGAAGATGACGCTGCT AGGCCTTCTTTAGTGTCTCTTGTA	Wu et al. 2012 [50]
*PAD4*	AT3G52430	GGTCGACGCTGCCATACTCAAACT AGAGAGATTGGTTTCCGAGCAGAG	Youssef et al. 2013 [51]
*RAR1*	AT5G51700	CGGCTCCTACTTCATCTCCAG AACATCGCAACATTTCCACCCTCT	Tornero et al. 2002 [52]
*SGT1b*	AT4G11260	CCCAAACCCAATGTCTCATCAG TCCACTTTCTTAGTCCCAACTTCT	Tör et al. 2002 [53]

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
