# Peer review of "BvcZR3 and BvHs1pro-1 Genes Pyramiding Enhanced Beet Cyst Nematode (Heterodera schachtii Schm.) Resistance in Oilseed Rape (Brassica napus L.)"

_ijms, 2019, doi:10.3390/ijms20071740_

Round 1
Reviewer 1 Report
Review
In the work entitled “BvcZR3 and BvHs1pro-1 genes pyramiding enhanced beet cyst nematode (Heterodera schachtii Schm.) resistance in oilseed rape (Brassica napus L.)” the authors perform in vitro and in vivo resistant test to beet cyst nematode making use of Brassica napus transgenic lines expressing cZR3 and Hs1 genes and hybridized lines expressing both genes.
I would like to highlight the positive points of the manuscript. The analysis of genes conferring resistance to diseases during plant growth is important and therefore, the present work brings new insights. The quality of pictures and graphs are very good and clear and illustrate well the results.
Major points to consider:
- General comment: The manuscript needs a major English review. There are very few sentences that do not contain grammar errors. Sometimes it´s quite hard to understand what the authors are trying to communicate. Discussion and materials and methods are the worse sections, hence requiring a deep English revision.
Minor points to consider:
- Page 1, line 12 – Please, review the english. Clearly indicate in each specie the genes are coming from.
- Page 1, line 12 – It´s not clear what is the translocation line neither B. procumbens.
- Page 1, line 15-16 – please rephrase the sentence. The work “respectively” doesn’t make sense here.
- Page 2, line 5 – Please review the english.
- Page 2, line 13 – eliminate “the”.
- Page 2, line 22 – Replace “cloning” by “cloned”.
- Page 2, line 24 – Remove “which”.
- Page 2, line 25 – Add “a” after “by”.
- Page 2, line 27 – Please replace “which contain” by “amid the presence of the”. Remove everything else after the “,” till the full stop or rephrase the sentence.
- Page 2, line 34 remove “has”.
- Page 2, line 35 – Replace “are consist” (this is a grammar error) by “contained the nucleotide-binding…”.
The libraries are of cDNA, not mRNA although they are obtained from mRNA. Therefore, Page 2, line 37 – Replace “exhibition” by “exhibiting”.
- Page 2, line 42 – Replace “with” by “to”.
- Page 2, line 43 – Add “that” after “shows”.
- Page 3, line 1 – Change to: “we hypothesized that…”.
- Page 3, line 3 – Do not introduce a paragraph here. Continue from the previous sentence.
- Page 3, line 4 – Remove “as explants”.
- Page 3, line 5 – Replace “cross” by “crossing”. Did the authors used heterozygous plants? That will not work. I think the authors wanted to write homozygous.
- Page 3, line 11-16 – Review the English. The grammar and sentence construction need review.
Figure 1 legend – Use Italic for all the scientific names.
- Page 3, line 18 and 22 – Replace “including” by “supplemented with”.
- Page 3, line 26-30 – Please indicate transformation percentages. I suggest a table with that data.
- Page 4, line 18 – Review the English, especially after (T0).
- Page 5, line 14 – Replace “For determination” by “In order to determine” and review the rest of the sentence.
- Page 5, lines 15 – Correct “in vitro”.
- Page 5, line 17 – Replace “including” by “containing”.
- Page 5, line 19 – Replace “in which” by “where”.
- Page 5, line 24 – Remove “The” and replace “normally” by “fully”.
- Page 5, line 31 – Add “that” after “suggested”.
- Page 8, line 12 – Replace “Genes were included” by “Gene list included”.
- Page 8, line 16 – Replace “the results showed the” by “the results showed that…”.
- Page 8, line 226 – Please rewrite. The grammar is incorrect.
- Page 13, line 24-26 – Please, review the english.
Figure 8 Legend – The authors write that significant difference was indicated by different letters (a, b) but the graph shows more letters such as d and e and bc. In fact, there is no reason to use different letters since they don´t differentiate nothing. One letter or symbol is more adequate to this kind of graphs.
- Page 10, line 9-11 – Review the english.
- Page 10, line 14 – Replace “exhibits and” by “exhibits an”
- Page 10, line 17 – Replace “was” by “were”.
- Page 10, line 19 – Review the English.
- Page 10, line 21 – Please, write the name of the author you are referring to and not just the number 21.
- Page 10, line 22-23 – Review the english.
- Page 10, line 31-37 – Please review the English. It´s very hard to follow.
- Page 11, line 9-11 – Review the english. It´s confusing.
- Page 12, line 14 and 22-25 – Review the English.
- Page 12, line 37 – Please never write “in addition to kanamycin”. It doesn´t make sense. Instead, say “supplemented with…”.
- Page 12, line 43 – Review the english.
- Page 13, lines 5 and 7– Review the english.
- Page 13, line 9 – One should read “leaves”.
- Page 13, line 13-14 – It´s impossible to understand the sentence.
Reviewer 2 Report
The authors have made attempts to transferred the Hs1pro-1 and cZR3 genes into oilseed rape using the hypocotyl explant as explants by Agrobacterium-tumefaciens mediated transformation method and pyramided the cZR3Hs1pro-1 genes by cross heterozygous transgenic cZR3 and Hs1 plants. The results are of interest for controlling beet cyst nematode. However, the current paper has not been well organized and poorly written as a scientific paper. There are numerous errors in both English grammar and spelling which were reduced the quality of the paper. The current paper should be critically revised and edited Some comments and suggestions may be useful for authors as below:
All scientific names of the plants, bacterial, genes should be made in italic, the format was not followed the style of the Journal. Given more explanations how did author design the vector?. Many sentences should be added references for the information confirmation. Some parts were duplicated, it should be removed. The discussion part must be significantly improved. References should be judiciously checked. (see comments and edits the attached file).

Author Response
Dear Reviewer:
Thank you for your comments concerning our manuscript entitled “BvcZR3 and BvHs1pro-1 genes pyramiding enhanced beet cyst nematode (Heterodera schachtii Schm.) resistance in oilseed rape (Brassica napus L.)” (Manuscript ID: ijms-463043). Those comments are all valuable and very helpful for revising and improving our paper, as well as the important guiding significance to our researches. We have studied comments carefully and have made correction which we hope meet with approval. Revised portion are highlighted in yellow in the paper. The main corrections in the paper and the responds to your comments are in the PDF file.

Round 2
Reviewer 2 Report
The authors have made a significant improvement in the revised version. I have some minor concerns and need to be confirmed by authors:
English editing needs to be improved before publishing, I have made some revisions which were directly highlighted in the pdf file of the manuscript, some names of bacteria should be made abbreviations throughout the text
Involving the vectors used in this study, the authors should confirm the copyright of material uses.

Author Response
Dear Reviewer:
Thank you for your comments about our revised manuscript entitled “BvcZR3 and BvHs1pro-1 genes pyramiding enhanced beet cyst nematode (Heterodera schachtii Schm.) resistance in oilseed rape (Brassica napus L.)” (ID: ijms-463043).
All revised portion have been clearly highlighted using “Track Changes” distinguished from the first revision.
The main corrections in the paper and the responds to the reviewer’s comments have been uploaded as a PDF file.
